# Membrane Transporters in *Citrus clementina* Fruit Juice-Derived Nanovesicles

**DOI:** 10.3390/ijms20246205

**Published:** 2019-12-09

**Authors:** Christopher Stanly, Maneea Moubarak, Immacolata Fiume, Lilla Turiák, Gabriella Pocsfalvi

**Affiliations:** 1EVs-MS Research Group, Institute of Biosciences and BioResources (IBBR), National Research Council, (CNR), 80131 Napoli, Italy; christopher.stanly@ibbr.cnr.it (C.S.); drmaneea1981@agr.dmu.edu.eg (M.M.); immacolata.fiume@ibbr.cnr.it (I.F.); 2MS Proteomics Research Group, Research Centre for Natural Sciences, Hungarian Academy Sciences, 1117 Budapest, Hungary; liliat7@gmail.com

**Keywords:** plant-derived vesicles, tetraspanins, transporters, intercellular communication, ATPases, mass spectrometry, bioinformatics, proteomics, biocargo, *Citrus clementina*, citrus

## Abstract

The cellular vesicle is a fluid-filled structure separated from the surrounding environment by a biological membrane. Here, we isolated nanovesicles (NVs) from the juice of clementines using a discontinuous density gradient ultracentrifugation method. To gain information about the protein content of vesicles, mass spectrometry-based organelle proteomics and bioinformatics were applied to the exosome-like vesicle fraction isolated in the 1 mol/L sucrose/D_2_O cushion. Analysis of 1018 identified proteins revealed a highly complex mixture of different intra, extracellular and artificially-formed vesicle populations. In particular, clathrin-coated vesicles were significantly expressed in this sample. Membrane transporters are significantly represented in clementines nanovesicles. We have found 162 proteins associated with the transport Gene Ontology term (GO: 0006810) which includes; 71 transmembrane transport related, 53 vesicle mediated and 50 intracellular transporters. Platellin-3 like carrier protein containing a Sec14 domain is known to have a role in plant-virus interaction and that is one of the most abundant proteins in our dataset. The presence of transmembrane transporters like ATPases, aquaporins, ATP Binding Cassette (ABC) transporters and tetraspanins, regulators of protein trafficking suggests that nanovesicles of clementines can actively interact with their environment in a controlled way.

## 1. Introduction

The cellular vesicle is a fluid-filled organelle separated from the surrounding environment by a biological membrane. The different kinds of vesicles, such as vacuoles, lysosomes, transport vesicles and secretory vesicles are important parts of a cell and can be classified based on their size, content such as coat protein, location and function. Plant vacuoles, for example, are large vesicles containing water, inorganic and organic molecules including enzymes, that besides other functions regulate the turgor pressure and the water level of cells [1]. In animal cells, lysosomes are packed with enzymes to digest proteins, polysaccharides, lipids, nucleic acids, damaged organelles, viruses and bacteria. Transport vesicles carry proteins and other molecules within the cell, for example from the endoplasmic reticulum to the Golgi apparatus and vice versa, or from one part of the Golgi to another [2]. Transport vesicles can be distinguished based on their protein coats that polymerize into cages to bend the membrane: clathrin-coated vesicles (CCVs), coat protein complex I (COPI) and coat protein complex II (COPII). The coats also have a function in molecular cargo selection and packing. Secretory vesicles are small membrane-surrounded packages in which nucleic acids, proteins, lipids, and metabolites are transported from organelles to a specific site on the cell membrane [3]. The existence of other small vesicles within cellular organelles like mitochondria, vacuoles, and multivesicular bodies, etc. are also described.

Cells ubiquitously secrete vesicles in all the three domains of life. Vesicles that are released from cells into the extracellular space are called extracellular vesicles (EVs) [4,5]. EVs play important roles in intercellular [5] and interkingdom communications [6,7] as well in immune responses [8]. EVs similar to intracellular vesicles are surrounded by a biological membrane and have a complex biocargo. The biogenesis of EVs is not fully understood, however, it is widely accepted that one type of vesicle called exosomes originates from the multivesicular bodies (MVBs) through the endocytic pathway. Secretion of vesicles in plants might be very similar to that of mammals, however no conclusive evidence has been obtained. Although transmission electron microscopy (TEM) images showed the presence of vesicle-like structures between the plasma membrane and cell wall of fungal hyphae more than 60 years ago [9], it took a long time to purify EV-like vesicles from plants. Regente et al. isolated small EV-like vesicles (20–200 nm in diameter), containing Rab11A from apoplastic washing fluids of sunflower seeds using the ultracentrifugation (UC) isolation method [10]. Recently, isolation methods employed in mammalian EV research have also been used to isolate vesicles from complex plant tissues like root homogenates and fruits. These preparations contain highly heterogeneous populations of both intra- and extracellular vesicles [6,7] and drew attention as alternative therapeutic solutions to mammalian cell-derived EVs [11]. For example, nanovesicles (NVs) isolated from grapes were shown to induce intestinal stem cells and protect against induced colitis in a mouse model. Raimondo et al. demonstrated that *Citrus limon*-derived nanovesicles inhibit cancer cell proliferation without affecting normal cells by activating TNF-alpha-related-apoptosis-inducing-ligand (TRAIL)-mediated apoptotic cell death [7]. Edible plant-derived vesicles are non-toxic, have tissue-specific targeting properties, and can be easily produced [12] making them promising as vectors for local and systemic delivery of various molecules. Fruit juices are one of the most popular sources to isolate edible plant-derived vesicles, as they are readily available by squeezing. By studying the protein biocargo of citrus fruit-derived vesicles, Pocsfalvi et al. have demonstrated the presence of highly expressed HSP70, HSP80, 14-3-3, G3PD and FBA6, PTL3 and clathirin proteins along with a large number of different enzymes and membrane transporters [13].

In this study, we show that the fruit juice of *Citrus clementina* contains nanovesicles packed with a complex protein biocargo. We found that the isolated vesicles contain different membrane transporters that may function in the movement of diverse molecular species across the membrane and thus may have an active role in cell–cell and interspecies communication.

## 2. Results and Discussion

### 2.1. Isolation of Nanovesicles (NVs) from Clementine Juice

Here, we isolated membrane-bound vesicles from the juice of the clementines using discontinuous density gradient UC. A schematic overview of the experimental workflow is shown in Figure 1 [8,14]. Briefly, juice was subjected to a series of low velocity centrifugation steps to remove sac cells, cellular debris, organelles and medium and large vesicles. Small vesicles containing pellet obtained after differential centrifugation was further purified and separated on 1 mol/L (M) and 2 M sucrose D_2_O cushions. The layer floating above the 1 M sucrose/D_2_O cushion (Figure 1A) with density similar to mammalian exosomes (1.15–1.19 g/mL) was collected, washed (refer to material and methods) and used for vesicle characterization and cargo analysis.

### 2.2. Characterization of Clementine Nanovesicles

For the characterization of particle size and concentration, nanoparticle tracking analysis (NTA) was used. NVs fraction showed a broad range size distribution from approximately 75 nm to 345 nm with an average size of 103.3 nm with standard deviation 52.3 nm (Figure 1B). Considering the complex character of fruit juice, we expected to purify a mixture of different intracellular and extracellular vesicle populations. In fact, NTA shows vesicle subpopulations at 75, 120, 155 nm (major peaks in the NTA) and at 235 and 345 nm (minor peaks) in the 1M fraction. Sodium dodecyl sulfate polyacrylamide gel electrophoresis (SDS-PAGE) analysis of the two vesicle fractions purified in the 1 M and 2 M sucrose/D_2_O cushions (Figure 1B) shows different profiles. The 1 M fraction reveals a highly complex pattern as it is generally observed in EV studies [13]. The yields determined after density cushion UC in the 1 M EV-like fraction are 1.16 × 10^12^ particles/L of fruit juice and 1.67 mg of NV proteins/L of fruit juice. This is at least 100 times higher than EV yields from mammalian cells and biological fluids.

### 2.3. The Protein Cargo of Citrus clementina Nanovesicles

Vesicles isolated in the 1 M sucrose/D_2_O fraction of *C. clementina* fruit juice were lysed by repeated freeze-thaw cycles in the presence of detergent and protein content was analyzed using mass spectrometry-based organelle proteomics. We identified 1018 proteins against the *C. clementina* UniProt database (31,274 entries) with log prob >3 values (Appendix A). A comparative study was performed to highlight similarities and differences between protein biocargo of *clementina* vesicles and existing datasets. Orthologous groups (OGs) of each identified protein (678 hits, in Appendix A) predicted by EggNOG mapper [15] were searched against (i) the OG accession codes published in EVpedia (27,517 hits, in Appendix A) [14] and (ii) four different citrus species (995 unique hits, in Appendix A) published recently [13]. The Venn diagram in Figure 1C shows the high overlap percentages found with both EVpedia (548 hits, 85%) and four citrus data sets (543 hit, 84%). The 83 clusters of orthologous groups (COGs) common with EVpedia but not present in the other citrus species studied is a unique feature of *C. clementine*-derived vesicles.

Most of the identified proteins were “uncharacterized”, therefore, in silico methods were used for their functional elucidation. Table 1 shows the 20 top-ranking proteins together with their functional annotation and identification counts in the EVpedia and 4-citrus data sets [13]. Most of the top-ranking proteins are frequently identified in EV related works as well they have been identified in other citrus species, like clathrin, patellin-3 like, heat shock, actins, ATPase and CDC48 proteins. Especially interesting are the two proteins associated to vesicle formation and transport: clathrin heavy chain and patellin-3 like protein. Clathrin coat protein is a key structural component of membrane budding and vesicle formation and was the first ranking protein in our data set. Patellin-3 like carrier protein is a plasma-membrane protein involved in vesicle and membrane-trafficking associated with cytokinesis and cell plate formation. These two proteins were found to be highly expressed in fruit-derived vesicles from other citrus species, too [13]. *Clementina*-derived vesicles also contain a variety of enzymes: carboxylase, mutase and ATPase, sucrose synthase, phosphorylase, etc. (Table 1, Appendix A). Since vesicles transport bioactive cargo within and between cells, enzymes may have a role in the mediation of activities both in local microenvironment and distant inter-organ communications.

### 2.4. Proteins Involved in Molecular Transport

Protein/lipid composition plays a crucial role in the formation of different vesicles [16]. Membrane proteins (MPs), besides being important structural elements of the vesicle membranes, also participate in specific functions such as cargo selection, fusion and interaction with environment. They are transporters, channels, enzymes, receptors, anchorage site and signal transductors. Many integral MPs are transmembrane proteins (TM), which span the lipid bilayer with portions exposed on both sides of the membrane. The presence of tetraspanins [17], and aquaporins [18] MPs has been described in the various vesicle types. In our dataset, we have found 162 proteins associated with transport Gene Ontology (GO) term GO: 0006810 (Appendix A) which includes 71 transmembrane transporters, 53 vesicle mediated transporters and 50 intracellular transporters (Figure 2A).

#### 2.4.1. Transmembrane Transporters in Clementine Vesicles

Transmembrane transporters are involved in the movement of ions, small molecules, macromolecules and cellular components across the lipid bilayer. Most of the identified transmembrane transporters in our dataset are ATPases (25 ATPases and 12 ABC subunits, Appendix A) and aquaporins (6 proteins, Appendix A). ATPases operate within the biological membranes moving water, sugars, amino acids, proton, cations and anions across the membranes. There are four major types of ATPases, namely P, V, F and ABC types. Our study revealed the presence of 10 vacuolar H^+^-ATPase (V-ATPase) subunits. V-ATPase is directly involved in the establishment and maintenance of the acidic pH of cells, endocytic and secretory organelles [19]. Besides their canonical functions, the roles of V-ATPases in membrane fusion, endocytosis and vesicular trafficking have also been reported [19]. In plant, electrochemical H^+^-gradient generated by V-ATPases energizes secondary transport of metabolites across the vacuolar membrane (tonoplast) and other endomembrane. In fact, tonoplast has been shown to be highly enriched in V-ATPases. Proteomics revealed the presence of prominent V-type ATPase subunits in tonoplast vesicle preparations [20]. One reasonable explanation of the high expression of V-ATPases in our *C. clementina* juice sac cells-derived vesicle sample, therefore, could be the presence of tonoplast vesicles. During the production of the juice, low-density small vesicles can easily have formed from the rupture and re-vesiculation of tonoplast and been purified by density gradient ultracentrifugation. The protein dataset revealed four P-type ATPases in *C. clementina*-derived vesicles. The P-type ATPases by translocating cations, heavy metals and lipids maintain the electrochemical gradients across cellular membranes. In fruit juice cells of *Citrus* species, the vacuole can be very acidic (pH 2). Recent work shows that for this hyper acidification a vacuolar proton-pumping P-ATPase complex could be responsible [19].

Plants have many more ATP Binding Cassette (ABC) transporters than mammalian cells [21]. ABC transporters are grouped into 8 subfamilies. In our samples, we have found 11 ABC transporter subunits from 6 different subfamilies A, B, C, E, F and G. While some ABC transporters have specific endogenous secondary metabolite substrates including lipids, carbohydrates, phytohormones, etc., others may transport many different chemically unrelated substances. Besides ATPases, the presence of aquaporin channel proteins were also characteristic of clementine vesicles.

Aquaporins by regulating the flux of water and other small solute molecules across membranes are not only important for plant physiology but also for the interaction between plants and the environment. Vesicles containing aquaporin have recently come into the focus of EV research [18]. Here, the first time we demonstrate that vesicles of clementina contain tip (tonoplast intrinsic protein) and pip (plasma membrane intrinsic proteins) types of aquaporins.

#### 2.4.2. Vesicle-Mediated Transport: Clathrin-Coated Vesicle (CCV), Coat Protein Complex I (COPI) and Coat Protein Complex II (COPII) Vesicles

Putative vesicle related transport proteins of clementina were identified by analyzing the proteins that are associated with transport GO terms (GO: 0006810) (Appendix A). Out of 162 proteins in our experimental data set associated with transport 53 were related to “vesicle-mediated transport” (GO: 0016192) (Figure 2B). Analysis of the dataset shows the presence of proteins related to all the three main types of intracellular vesicles: CCVs, COPI and COPII vesicles. CCV related proteins were clathrin light- and heavy-chain coat proteins as well different subunits of AP-1, AP-2 and AP-4 complexes. Several proteins of COPI transport vesicles were also identified, including 7 coatomer complex subunits (alpha-1, beta-1, delta, epsilon-1, zeta-1, gamma-1 and beta-2). COPII vesicle-related proteins including 11 sec proteins and two GTP binding proteins SARA1 were also present in *clementina* vesicle. Proteomic data confirms our previous observations [13] that the clementina similarly to bitter orange, grapefruit, lemon and sweet orange-derived vesicles contains vesicular transport-related proteins characteristic to CCVs, COPI and COPII trafficking vesicles and EVs.

#### 2.4.3. Regulators of Trafficking: The Tetraspanins

Tetraspanins are the main structural elements of the membrane microdomains (referred to as tetraspanin-enriched microdomains) and regulators of protein trafficking. Numerous physiological and pathological roles have been associated to tetraspanins. Recently, the role of tetraspanins and tetraspanin-associated proteins in the biogenesis of EVs has been proposed. Tetraspanins CD9, CD63, and CD81 are frequently present in EV samples and are often used as EV biomarkers. Plant tetraspanins have been overlooked for a long time but research is emerging thanks to their newly discovered roles in intercellular and interspecies communication [22]. Seventeen tetraspanin genes have been described in the model plant *Arabidopsis*. Here we were interested in whether tetraspanins are also a structural part of the plant derived vesicles. Blast sequence similarity searching was applied to identify *Arabidopsis* tetraspanin “homologues” (20 proteins) in the proteome of *C. clementine* (Appendix A). The 13 homologues identified were then searched for in our vesicle related protein data set. In clementine-derived nanovesicles (1 M fraction) two tetraspanins, tetraspanin-3-like and tetraspanin-8-like were identified. Both tetraspanins are known to be present in *Arabidopsis* female reproductive organ (pistil or carpel). Moreover, proteomic analysis of plasmodesmata, the channels that link adjacent cells in plant tissues isolated from populus cell suspension, has also shown enrichment of the same tetraspanin-3 and 8 [23]. Based on this data, we can speculate that the vesicle population expressing these tetraspanins in our sample can be those that formed from the membrane of the plasmodesmata.

## 3. Materials and Methods

### 3.1. Fruit Material and Vesicle Isolation

Fruits of *Citrus clementina*, a seedless Italian cultivar produced around Naples collected in Naples, Italy on 24 February 2018. 20 pieces of fresh fruits were gently squeezed with the help of a glass lemon/orange squeezer and filtered using filter paper to obtain about 800 mL of fruit juice. The isolation procedure was performed in duplicates at room temperature. For the second isolation, 400 mL of fruit juice was used. Protease inhibitor cocktail containing 0.5 mL, 1 mg/mL Leupeptine, 2.5 mL 100 mM Phenylmethylsulfonyl fluoride (PMSF) and 1.6 mL 1 M sodium azide were added to every L of sample. Vesicles were isolated by differential UC following the procedure described by Stanly et al. [24] Briefly, low-velocity centrifugation steps were performed at 400× *g* and 800× *g* for 20 min at 22 °C to eliminate cells and cell debris, and at 15,000× *g* for 20 min at 22 °C to collect the fraction enriched in microvesicles. The supernatant was ultracentrifuged at 150,000× *g* for 60 min at 4 °C using a Type 70 Ti Beckman rotor. The obtained pellet was resuspended in 28 mL of 20 mM Tris-HCl (pH 8.5), under-layered by two cushions composed of 1 and 2 mol/L sucrose prepared in Tris-HCl/D_2_O, and centrifuged at 110,000× *g* for 180 min at 4 °C using SW 32 Ti rotor. The resulting vesicle layers were collected separately, washed twice in 20 mM Tris-HCl and collected by centrifugation at 110,000× *g* for 60 min at 4 °C using a SW 32 Ti rotor. The pellet was resuspended in 100–150 µL 20 mM Tris-HCl buffer and micro bicinchoninic acid (BCA) assay was performed to determine the protein concentration.

### 3.2. Physiochemical Characterization of Different Vesicle Populations

#### 3.2.1. Nanoparticle Tracking Analysis

Particle size distribution was determined by NTA using a NanoSight NS300 system (Malvern Technologies, Malvern, UK) configured with a 488 nm laser and a high sensitivity scientific complementary metal-oxide semiconductor (CMOS) camera. Sample was diluted 1:200 in particle-free PBS. Samples were analyzed under constant flow conditions (flow rate = 50) at 25 °C. Data were analyzed using NTA 3.1.54 software with a detection threshold of 5.

#### 3.2.2. Proteomics and Data analysis

##### SDS-PAGE Analysis

The quality of the vesicle samples and the reproducibility of the isolation was checked using sodium dodecyl sulfate polyacrylamide gel electrophoresis (SDS-PAGE). Samples (20 μg protein measured by micro BCA assay) were used as described in paragraph “Lysis of the Vesicles and Shotgun Proteomics Analysis” and electrophoretically separated under reducing conditions on a precast Novex Bolt 4%–12% Bis-Tris Plus gel using Bolt (3-(N-morpholino)propanesulfonic acid) MOPS SDS running buffer (Life Technologies, Carlsbad, CA, USA) according to the manufacturer’s instructions and stained with colloidal Coomassie blue.

##### Lysis of the Vesicles and Shotgun Proteomics Analysis

For in-solution proteomics analysis, samples (100 μg protein measured by micro Bicinchoninic acid assay) were resuspended in 0.2% RapiGest SF (Waters Corp., Milford, MA, USA) and vesicles were lysed using five freeze-thaw cycles consisting of alternative freezing the samples for 5 min in a dry ice–ethanol slurry followed by thawing them in a bath of water under sonication at room temperature for 5 min. Proteins were reduced using 5 mM DL-dithiothreitol (Sigma-Aldrich, Saint Louis, MO, USA,), alkylated using 15 mM iodoacetamide (Sigma-Aldrich, Saint Louis, MO, USA), and digested using MS grade trypsin (Pierce, Thermo Sci. Rockford, IL, USA) as described previously [24]. Samples were vacuum dried and solubilized in 5% acetonitrile and 0.5% formic acid (liquid chromatography–mass spectrometry (LC–MS) grade, Thermo Sci. Rockford, IL, USA) prior to nano high-performance liquid chromatography–tandem mass spectrometry (nano-HPLC-MS/MS) analysis.

##### Nano Liquid Chromatography-Electrospray Ionization–Tandem Mass Spectrometry (NanoLC-ESI-MS/MS)

Shotgun proteomics analysis was carried out on 5 μg of tryptic digest using a Dionex Ultimate 3000 nanoRSLC (Dionex, Sunnyvale, CA, USA) coupled to a Bruker Maxis II mass spectrometer (Bruker Daltonics GmbH, Bremen, Germany) via CaptiveSpray nanobooster ionsource. Tryptic digest samples were desalted on an Acclaim PepMap100 C-18 trap column (100 μm × 20 mm, Thermo Scientific, Sunnyvale, CA, USA) using 0.1% TFA for 8 min at a flow rate of 5 μL/min and separated on the ACQUITY UPLC M-Class Peptide BEH C18 column (130 Å, 1.7 μm, 75 μm × 250 mm, Waters, Milford, MA, USA) at 300 nL/min flow rate, 48 °C column temperature. Solvent A was 0.1% formic acid, solvent B was acetonitrile, 0.1% formic acid and a linear gradient from 4% B to 50% B in 90 min was used. Mass spectrometer was operated in the data dependent mode using a fix cycle time of 2.5 s. MS spectra was acquired at 3 Hz, while MS/MS spectra at 4 or 16 Hz depending on the intensity of the precursor ion. Singly charged species were excluded from the anaylsis.

##### Protein Identification, Quantification and Statistical Analysis

Raw data files were processed using the Compass Data Analysis software (Bruker, Bremen, Germany). Proteins were identified using the Byonicv.3.4.0 software against the *Citrus clementina* UniProt database (31,274 entries). The search criteria were the following: 20 ppm precursor and fragment mass tolerance. Two missed cleavages were allowed, and the following modifications were set: carbamidomethylation on cysteine as fixed modification, while methionine oxidation and asparagine and glutamin deamidation as variable modifications. Protein identifications were validated by the Percolator algorithm [25] false discovery rate was <1%. Protein data is available at EV-Track under ID: EV190071.

##### Bioinformatics

Tag distributions of OmicsBox, version 1.1.74: 1018 sequences were blasted using QBlast service against the National Center for Biotechnology Information (NCBI) public databases using taxonomy filter green plants (taxa: 33,090, Viridiplantae), number of blast hits 20 and expectation value 1.0 × 10^−3^. The InterPro domain searches were performed using the public European Molecular Biology Lab-European Bioiformatics Institute (EMBL-EBI) InterPro web services to seen sequences against Interpro’s signatures (CDD, HAMAP, HMMpfam, HMMPIR, Fprintscan, BlastproDom). All signatures generated interpro results. All together 949 sequences were annotated and 939 were mapped against exclusively created GO-annotated proteins to obtain functional labels of GO-associated and Uniprot’s ID mapping. Orthology assignment and COGs annotation were performed by EggNOg Mapper version 5.4.1 [15]. EggNOG OGs were used to compare protein datasets between different taxa and EVpedia [14,26].

## 4. Conclusions

In this study, exosome-like nano-sized vesicles have been isolated with a high yield (1.6 mg/L) from the fruit juice of *C. clementine*. NTA and proteomic analysis show a heterogeneous mixture of different vesicle subpopulations. Proteomic and bioinformatic studies revealed that all the three major classes of coated vesicles, i.e., CCV, COPI and COPII, distinct in their protein compositions as well as tonoplast, apoplast and plasmodesmata-derived nanovesicles, present in this preparation. Amongst the top-ranking proteins are clathrin heavy chains associated with vesicle formation and patellin-3 like carrier proteins associated with cytokinesis-related cell division. Besides the natural intra- and extracellular vesicles, we propose that plasma and vacuole membranes as well as the limiting membrane of the plasmodesmata can give rise to the formation of artificial vesicles during the preparation of the fruit juice and isolation of the vesicles. Vesicle samples isolated from edible-plants foods such as whole organs, tissues, fruit juice, etc. therefore inherently contain a complex mixture of natural and artificial vesicles. The presence of different membrane transporters like aquaporin, ATPases and ABC transporters as well as tetraspanin regulators implies their possible participation in translocation and transportation of various substances between the vesicle interior and environment. However, to gain a better insight, the results of this study need to be integrated with biochemical, physiological and functional assays to establish that the identified transporters are indeed functional. Understanding the molecular components of the plant food-derived vesicles, including enzymes and transporters, is expected to widen our knowledge on how vesicles can be used as vectors for local and systemic molecular delivery. More generally, this information can be of value in the exploitation of edible plant-derived vesicles in biotechnology and biomedicine.

## Figures and Tables

**Figure 1 ijms-20-06205-f001:**
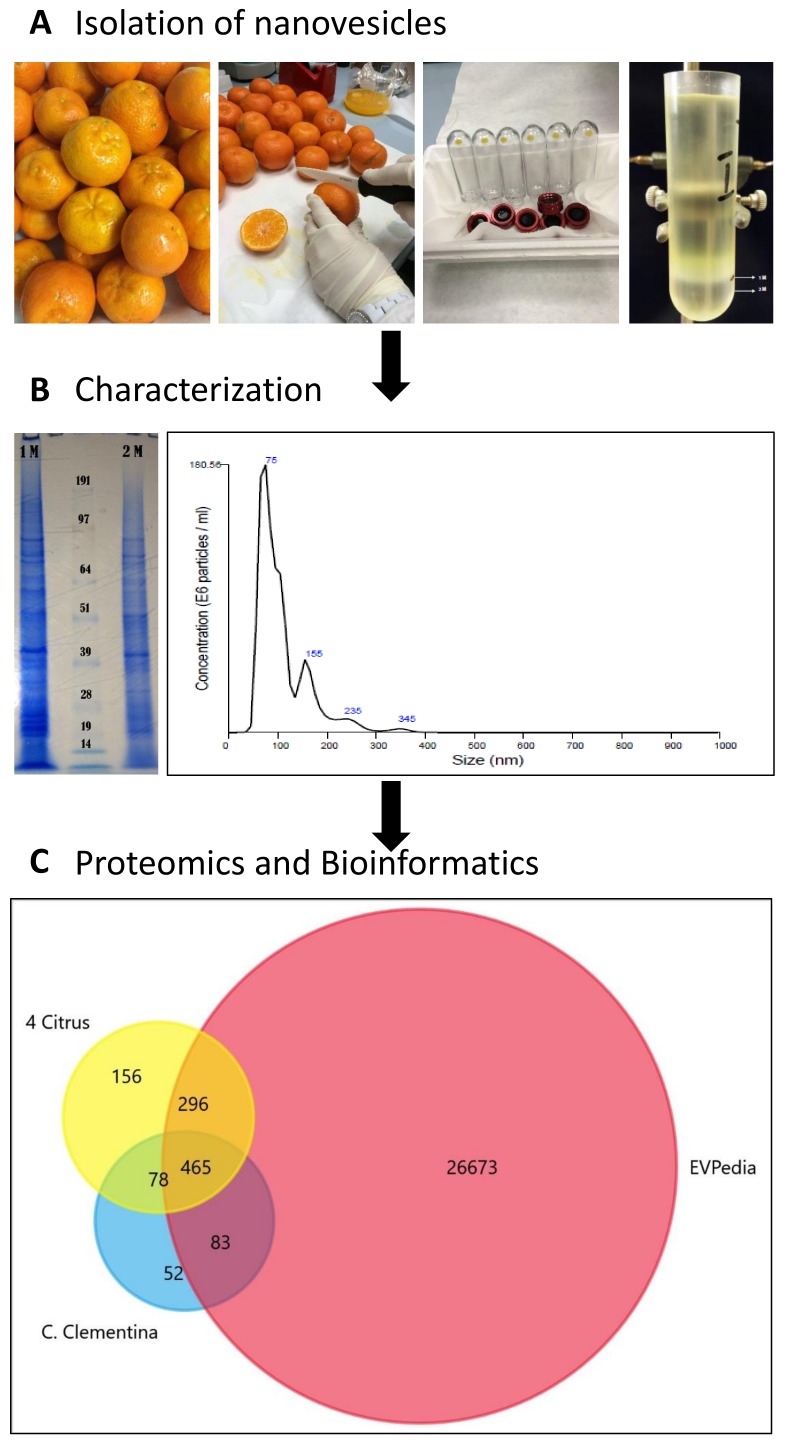
Schematic chart of the experimental work performed to isolate, characterize and analyze *C. clementine* fruit juice-derived exosome-like nanovesicles. (**A**) Lower left image shows the pellets obtained after diffferential ultracentrifugation (UC) lower right image shows the separation obtained by sucrose/D_2_O double cushion UC. The vesicles floating above the 1M sucrose/D_2_O cushion were found to be similar in density to mammalian extracellular vesicles. (**B**) sodium dodecyl sulfate polyacrylamide gel electrophoresis (SDS-PAGE) protein profiles (right) and of vesicle populations in the 1 M and 2 M sucrose/D_2_O cushions and the particle-size distributions of vesicles isolated in the 1M sucrose/D_2_O cushion and measured using nanoparticle tracking analysis (NTA). (**C**). Venn diagram generated by FunRich software [1] shows the numbers of unique and common Orthologous Groups (OGs) of the identified protein. OGs of *Citrus clementina* (azure) were compared to four citrus species (*C. sinensis*, *C. limon*, *C. paradise* and *C. aurantium*) (yellow) [13] and EVpedia (red) [14].

**Figure 2 ijms-20-06205-f002:**
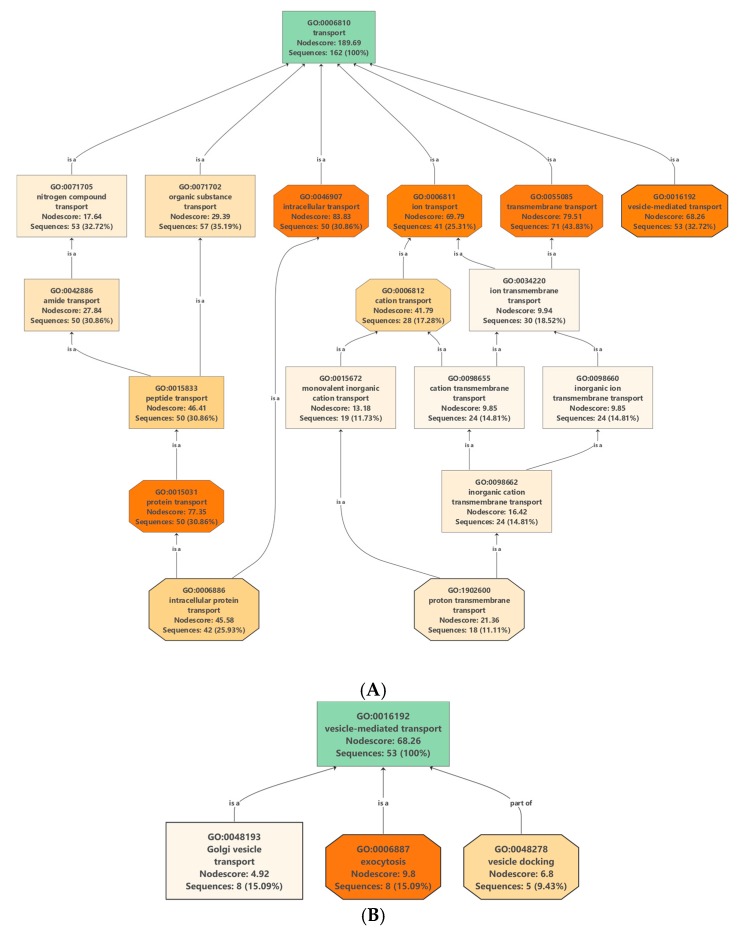
Enrichment and distribution of (**A**) transport-related Gene Ontology (GO): 0006810 and (**B**) vesicle mediated transport-related GO: 16192 terms in the protein data set of nanovesicles isolated from the juice of *C. clementine*.

**Table 1 ijms-20-06205-t001:** List of 20 top-ranking proteins identified in exosome-like nanovesicles containing fraction of *Citrus clementina*. UniProt accession number indicate the accession number from the UniProt database; UniProt description is the protein identification from the UniProt database; Byonic scores indicate the scores obtained from the byonic software; cluster of orthologous groups (COGs) accession number obtained by EggNOG mapper; EggNOG description was retrieved from the EggNOG database; EVpedia ID count indicates the frequency score published in the EVpedia database; 4 citrus indicates the presence or absence of protein in our previous study [13].

Accession UniProt	UniProt Description	MW [kDa]	pI	Scores	COG	EggNOG Description	EVpedia	4 Citrus
V4V9V7	Clathrin heavy chain	192.8	5.3	814.0	ENOG410XPH1	Clathrin heavy chain	451	Y
V4T403	Uncharacterized protein	64.6	4.9	906.9	ENOG410XRSQ	Carrier protein, Patellin-3 like	44	Y
V4TVM6	Uncharacterized protein	101.2	5.8	740.5	COG0542	Heat shock protein	61	Y
V4VRS8	Uncharacterized protein	110.9	6.1	646.6	COG2352	Phosphoenolpyruvate carboxylase	10	N
V4UFD6	Uncharacterized protein	71.0	5.1	860.1	COG0443	Heat shock cognate 70 kDa	608	Y
V4SV61	Uncharacterized protein	61.0	5.7	880.4	COG0696	Phosphoglycerate mutase	15	Y
V4SQM8	Uncharacterized protein	154.8	4.7	621.9	n.d.	Unknown	n.d.	N
V4WGV2	Uncharacterized protein	80.0	5.0	730.9	COG0326	Heat shock protein	557	Y
V4SQ92	Uncharacterized protein	41.7	5.3	757.5	COG5277	Actins	592	Y
V4ULL2	Uncharacterized protein	94.1	5.1	659.3	COG0443	Heat-shock 70 kDa protein	608	Y
V4SU87	Plasma membrane ATPase	105.2	6.1	810.6	COG0474	Plasma membrane ATPase	378	Y
V4TDB5	Uncharacterized protein	68.6	5.3	710.6	COG1155	V-type proton ATPase catalytic subunit	21	Y
V4U840	Uncharacterized protein	93.8	5.9	759.8	COG0480	Elongation factor	500	Y
V4T1U7	Uncharacterized protein	47.1	6.0	898.2	COG0446	Monodehydroascorbate reductase	112	Y
V4W5U2	Uncharacterized protein	63.3	5.5	696.8	COG0033	Phosphoglucomutase	217	Y
V4RUA3	Alpha-1,4 glucan phosphorylase	95.2	7.0	780.2	COG0058	Phosphorylase	297	Y
V4TVW5	Sucrose synthase	92.6	6.0	660.8	COG0438	Sucrose-cleaving enzyme	154	Y
V4UJC4	Uncharacterized protein	90.2	6.3	649.7	COG0339	Oligopeptidase	142	Y
V4TCA6	Uncharacterized protein	252.7	6.0	601.6	COG0439	Acetyl-CoA Carboxylase	159	Y
V4SL33	Uncharacterized protein	89.4	5.1	575.2	COG0464	Cell-division cycle protein 48	479	Y

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
