# Peer review of "Membrane Transporters in Citrus clementina Fruit Juice-Derived Nanovesicles"

_ijms, 2019, doi:10.3390/ijms20246205_

Round 1

Reviewer 1 Report

The work provides a protocol for isolating nanovesicles from Citrus clementine fruit juice. The vesicle protein inventory is analyzed by LC-MS using tryptic digests with partial identification of the proteins and descriptions coming from a variety of bioinformatics tools.

The work is well documented and summarized in Table 1 and Figure 2. The two most highly expressed proteins are identified and their putative roles in plant transport and vesicle integrity described.

The significance of the study is not explicit in the authors manuscript. Although it positions the work in relation to similar studies, the large number of uncharacterized proteins makes the next steps harder to see. The authors should point out how their results aid in understanding vesicle transport and/or regulation of molecular traficking in plant cells.  Or perhaps the results are expected to bear on how vesicles can be used as promising vectors for local and systemic molecular delivery. A delineation of the path forward would be appreciated by the less sophisticated reader.

Some minor edits:

Section 2. Results should be Section 3. Results (line 157).

Lines 65 and 66 (after Table 1) contain a dangling sentence: "Regarding the roles of aquaporins in plant EVs, there no studies that demonstrated." This should be completed.

Author Response

Dear Editor,

We would like to take this opportunity to thank the reviewer1 for his/her constructive comments. The article has been revised according to the suggestions and required changes have been made. We hope that the article now meets the publishing criteria and that it would be accepted for publication.

Point-by-point response to Reviewer 1

Response to the overall comment:

To get the significance of the study more explicitly said in the manuscript we have made three major modifications:

The abstract of the manuscript was updated to highlight the 3 main research findings: complexity of the different vesicle subpopulations in fruit juice the two highly expressed vesicle-related proteins (clathrin and patellin-3) the presence and roles of different membrane transporters The conclusion part of the manuscript was also re-elaborated. We have tried to underline better how this kind of basic study can help to advance our understanding in the exploitation of edible plant-derived vesicles. We have changed the title (also based on the suggestion of reviewer 2)

We are confident that the revised version of our manuscript emphasizes our main findings in a better way. We agree that the work on C. clementina is still hampered by the limited numbers of characterized proteins (the reason for the use of bioinformatic tools). We also hope that some of the proteins identified in this study will be structurally and functionally characterized by further studies.

Editing minor errors (answers are highlighted by green):

Section 2. Results should be Section 3. Results (line 157).

We have changed the order of the sections as suggested

Lines 65 and 66 (after Table 1) contain a dangling sentence: "Regarding the roles of aquaporins in plant EVs, there no studies that demonstrated." This should be completed.

This error has also been corrected in the revised manuscript.

-------------------------------------------------------------------------------------

Reviewer 2 Report

1) Title of the submitted Manuscript requires revision.

Nanovesicles from juice sac cells of Citrus clementina:

3 A quick glance on membrane transporters

There are no any biochemical/physiological/functional assays for transporters provided in the text.

Secondly, the Authors mention high enrichment of isolated vesicles with actin, PEPC, cell division cycle which are typical cytoplasmic proteins. Then heat shock prions etc.

What was the size of the vesicles? The quality of figure 1b does not allow to make any estimates at all.

2) Quality of submitted figures (figure 1a, 1b, 1a) is below any level of appreciation, they are not visible in pdf and require complete revision.

3) The part from results 3.1

“158 3.1. Isolation of nanovesicles from clementine juice

159 Citrus is an evergreen, subtropical tree widely cultivated for their juicy edible fruits. Plants in the

160 Citrus genus produce citrus fruits, including important crops such as oranges, lemons, grapefruits,

161 pomelos, limes and clementines. Citrus taxonomy is complex and the exact number of natural species

162 is unclear, as many of the named species are hybrids clonally propagated through seeds. Clementine

163 (Citrus clementina) is a hybrid between a willow leaf mandarin orange (C. deliciosa) and a sweet orange

164 (C. sinensis). Here, we isolated membrane-bound vesicles from the juice of the clementines using density

165 gradient ultracentrifugation (UC), one of the most widely used methods for the isolation of membrane

166 bound vesicles from various sources [8] [14] Briefly, juice was subjected to a series of low velocity

167 centrifugation steps to remove sac-cells, cellular debris, organelles and medium and large vesicles.

168 Small vesicles containing pellet obtained after differential UC was further purified and separated using

169 double sucrose D2O cushions. The layer floating above the 1 mol/L (M) sucrose/D2O cushion (Figure

170 1A) with density similar to mammalian exosomes (1.15 to 1.1.9 g/mL) was collected and washed. The

171 purified nanovesicles (NVs) containing pellet was resuspended in buffer (Refer to material and

172 methods) and used for vesicle characterization and cargo analysis.”

should be divided between methods and introduction, here are not the results but the introduction about the plants/trees, then part of methods.

4) 83 3.4.3. Regulators of trafficking: The Tetraspanins

84 Tetraspanins are the main structural elements the membrane microdomains (referred to as

85 tetraspanin-enriched microdomains), regulators of protein trafficking.

Any references? What are the other proteins involved in trafficking? Why the other proteins are not mentioned?

5) The description of methods is poor. What was the used cultivar of fruits? Please, provide details for squeezing procedure, time of year when the fruits were collected.

6) What was the number of replicates for the experiments? Number of separate plants for the 30 pieces of fruits? Were the bits of tissues removed? Temperature of the procedure?

7) ATPases operate within

35 the biological membranes with the purpose of moving water, sugars, amino acids, proton, cations

36 and anions across the membranes. There are four major types of ATPases, namely P, V, F and ABC

37 types.

Not correct. ATPase do not operate with the purpose to “move water, sugars etc.”. It’s too much indirect statement.

ABC transporters are typically considered separately from P-type, V-type ATPases. ATPases in mitochondria and chloroplasts are different types of ATPases.

8) 12 Abstract: Cellular vesicle is a fluid filled organelle separated from the surrounding environment by a

13 biological membrane.

There could be plenty of artificially formed vesicles during destruction of material, then could be two membranes etc. Essential parts of the text should be rewritten.

9) Plant vacuoles, for example, are large vesicles containing

35 water, inorganic and organic molecules including enzymes, that besides other functions regulate the

36 turgor pressure and the water level of cells [1].

Vacuole in plants is typically considered as an important pant cell organelle, not a vesicle, it is not regulating “water level of cells” but mostly passively participates in control of water potential of plant cells.

10) 38 bacteria. As the name suggests, transport vesicles carry proteins and other molecules within the cell,

39 for example from endoplasmic reticulum to Golgi apparatus and vice versa or from one part of Golgi

40 to another [2].

Is it better to write a literature text with the phrases “as the name suggests’; the descriptions are not succinct as required for a reasonable scientific text.

11) Table 1 is the best seen since the figures are not visible. How can the Authors explain that the vesicles enriched with plasma membrane and vacuolar ATPases?

Just the destruction of the cells and artificial formation of closed vesicles from plasma membrane and of vacuolar membrane.

12) Please, provide comparison of the vesicles by your method with vesicles isolated under cold in any type of description (size, electrophoretic properties etc.).

13) The Reviewer is sorry to say but the parts of the text are rather about nothing without clear description and statements. E.g., again just simple general phrases as below. Pls, brush up all the text making it clear and concentrated.

3.4. Proteins involved in molecular transport

18 Analysis of different types of vesicles shows the protein/lipid composition plays a crucial role in

19 their formation [17]. Membrane proteins (MPs), besides being an important structural element of the

20 vesicle membranes, are also responsible for carrying out specific functions such as cargo selection

21 and fusion and allowing the vesicle to interact with its environment in a controlled way. MPs that are

22 prevalently responsible for the interaction between vesicle and environment are transporters,

23 channels, enzymes, receptors, anchorage site and signal transductors. Many integral MPs are

24 transmembrane proteins (TM), which span the lipid bilayer with portions exposed on both sides of

25 the membrane. The presence of MPs such as tetraspanins [18], and aquaporins [19] has been described

26 in the various vesicle types.

14) The P-type ATPases are

49 known to form a phosphorylated intermediate state during their ion transport cycle [22]. They

maintain the electro-chemical gradients across cellular membranes, 50 by translocating cations, heavy

51 metals and lipids. In fruit juice cells of Citrus species, the vacuole can be very acidic (pH 2). Recent

52 work shows that for this hyper acidification a vacuolar proton-pumping P-ATPase complex could be

53 responsible [20].

Reference [22] is 210 22. Termini, C.M.; Gillette, J.M. Tetraspanins Function as Regulators of Cellular Signaling. Front.

211 Cell Dev. Biol. 2017, 5, 34–34.

The text is again simply too description and not precise.

15) Technically, numbering of lines starts from page 7 repeatedly, the same with page numbering.

16) 54 Plants have many more ATP Binding Cassette (ABC) transporters than mammalian cells. ABC

55 transporters are grouped into 8 subfamilies. In our samples, we have found 11 ABC transporter

56 subunits from 6 subfamilies A, B, C, E, F and G.

References?

17) Aquaporins care channel proteins regulating the flux of

61 water and other various small solutes across membranes. Aquaporins are important not only for the

62 plant physiology in the maintenance of water homeostasis but also in the interaction between plant

63 and environment. Vesicles containing aquaporin are recently get into the focus of EV research.

Language is not good, pls, check all the text.

“Aquaporins care channel proteins…” is nonsense.

Interaction between plant and environment is not related to plant physiology?

18) All the text with plenty of potentially good and valuable data should be completely revised and restructured/reconsidered. So far here is a very shallow approach.

Author Response

Response to Reviewer 2

The authors would like to thank Reviewer 2 for his critical and constructive observations. We have considered all the suggestions and based on that we have completely revised the manuscript. We believe that the manuscript in its present form better shows the results we have obtained in the analysis of clementines vesicles.

Reviewer 2

Open Review

Comments and Suggestions for Authors

1) Title of the submitted Manuscript requires revision. Nanovesicles from juice sac cells of Citrus clementina:

3 A quick glance on membrane transporters

The title has been changed to “Membrane transporters in Citrus clementina fruit juice-derived nanovesicles” We hope that our reviewers will find that the new title is more appropriate and reflects more the content of the manuscript.

There are no any biochemical/physiological/functional assays for transporters provided in the text.

Yes, as far as we are aware, so far there are no biochemical, physiological and functional assays performed on transporters in clementine-derived nanovesicles and it was outside of the scope of the present study. But we completely agree with this criticism and in the future, we are indeed interested to set-up and perform vesicular transport and uptake assays that could effectively show if the identified transporters are functional or not.

In addition, we also inserted the following paragraph in the Discussion part to underline the importance of the functional studies:

“The presence of different membrane transporters like aquaporin, ATPases and ABC transporters as well as tetraspanin regulators implies their possible role in translocation and transportation of various substances between the vesicle interior and environment. Although, to have a better picture, the results of this study needs to be integrated with biochemical, physiological and functional assays to establish that the above-identified transporters are indeed functional. Understanding the molecular components of the plant food-derived vesicles, including enzymes and transporters, is expected to widen our knowledge on how vesicles can be used as vectors for local and systemic molecular delivery. More generally, this information can be of value in the exploitation of edible plant-derived vesicles in biotechnology and biomedicine.”

Secondly, the Authors mention high enrichment of isolated vesicles with actin, PEPC, cell division cycle which are typical cytoplasmic proteins. Then heat shock prions etc.

What was the size of the vesicles? The quality of figure 1b does not allow to make any estimates at all.

Based on NTA measurements clementine vesicles (1M fraction) contains five discrete populations at 75, 120, 155 (major peaks) and 235 and 345 nm (minor peaks). Also, we cannot exclude the presence of co-purifying proteins of the fruit juice. On the other hand, it is not unusual to see cytoplasmic proteins in the vesicles. In fact, the inner liquid core of vesicles including EVs is known to contain cytoplasmic proteins, including actin and HSPs (see Vesiclepedia, EVpedia databases).

Regarding the quality of the image: for the revised manuscript we improved the quality of the images and rearranged Figure 1.

2) Quality of submitted figures (figure 1a, 1b, 1a) is below any level of appreciation, they are not visible in pdf and require complete revision.

Figure 1 was re-designed. It is now showing the workflow of the study and includes the pictures (that are provided separately in high definition) related to the isolation, characterization and proteomics and bioinformatics. We hope that the new figure will meet the required standards of the journal.

3) The part from results 3.1

“158 3.1. Isolation of nanovesicles from clementine juice

159 Citrus is an evergreen, subtropical tree widely cultivated for their juicy edible fruits. Plants in the

160 Citrus genus produce citrus fruits, including important crops such as oranges, lemons, grapefruits,

161 pomelos, limes and clementines. Citrus taxonomy is complex and the exact number of natural species

162 is unclear, as many of the named species are hybrids clonally propagated through seeds. Clementine

163 (Citrus clementina) is a hybrid between a willow leaf mandarin orange (C. deliciosa) and a sweet orange

164 (C. sinensis). Here, we isolated membrane-bound vesicles from the juice of the clementines using density

165 gradient ultracentrifugation (UC), one of the most widely used methods for the isolation of membrane

166 bound vesicles from various sources [8] [14] Briefly, juice was subjected to a series of low velocity

167 centrifugation steps to remove sac-cells, cellular debris, organelles and medium and large vesicles.

168 Small vesicles containing pellet obtained after differential UC was further purified and separated using

169 double sucrose D2O cushions. The layer floating above the 1 mol/L (M) sucrose/D2O cushion (Figure

170 1A) with density similar to mammalian exosomes (1.15 to 1.1.9 g/mL) was collected and washed. The

171 purified nanovesicles (NVs) containing pellet was resuspended in buffer (Refer to material and

172 methods) and used for vesicle characterization and cargo analysis.”

should be divided between methods and introduction, here are not the results but the introduction about the plants/trees, then part of methods.

In the revised manuscript, we deleted some parts (like the one that describes citrus plant) but keep a brief and less descriptive description of the isolation process.

4) 83 3.4.3. Regulators of trafficking: The Tetraspanins

84 Tetraspanins are the main structural elements the membrane microdomains (referred to as

85 tetraspanin-enriched microdomains), regulators of protein trafficking.

Any references? What are the other proteins involved in trafficking? Why the other proteins are not mentioned?

Other proteins involved in protein trafficking are described in a separate paragraph 2.4.1. Transmembrane transporters in clementine vesicles. Tetraspanins as regulatory proteins of trafficking that we describe in paragraph 2.4.1. References 22 and 23 describes tetraspanins in mammalian and plants, respectively.

5) The description of methods is poor. What was the used cultivar of fruits? Please, provide details for squeezing procedure, time of year when the fruits were collected.

In the revised manuscript we improved the description of the methodology. Cultivar, date of the collection and squeezing method were added.

6) What was the number of replicates for the experiments? Number of separate plants for the 30 pieces of fruits? Were the bits of tissues removed? Temperature of the procedure?

We performed the isolation experiments twice and it is now stated in the manuscript. Bits of tissues were removed by gravity (paper) filtration. Temperature of the vesicle isolation, including filtration is now described in the text.

7) ATPases operate within

35 the biological membranes with the purpose of moving water, sugars, amino acids, proton, cations

36 and anions across the membranes. There are four major types of ATPases, namely P, V, F and ABC

37 types.

Not correct. ATPase do not operate with the purpose to “move water, sugars etc.”. It’s too much indirect statement.

We have changed this.

ABC transporters are typically considered separately from P-type, V-type ATPases. ATPases in mitochondria and chloroplasts are different types of ATPases.

ABC transporters have ATPase domains and frequently considered as a class of ATPases.

8) 12 Abstract: Cellular vesicle is a fluid filled organelle separated from the surrounding environment by a

13 biological membrane.

There could be plenty of artificially formed vesicles during destruction of material, then could be two membranes etc. Essential parts of the text should be rewritten.

We have rewritten the abstract. The new version of the abstract summarizes our results and not the experiments.

The intracellular and extracellular vesicles naturally exist in the cells and its surrounding environment. We completely agree that during the isolation procedure, there is a possibility that vesicle like structures are also artificially formed and it is stated in the manuscript. Since, these structures have the same or similar biomembrane to the natural vesicles and/or plasma membrane they originate, they can also be active. Nevertheless, at this stage we are unable to distinguish between the two types.

9) Plant vacuoles, for example, are large vesicles containing

35 water, inorganic and organic molecules including enzymes, that besides other functions regulate the

36 turgor pressure and the water level of cells [1].

Vacuole in plants is typically considered as an important pant cell organelle, not a vesicle, it is not regulating “water level of cells” but mostly passively participates in control of water potential of plant cells.

We agree on this. Indeed vacuole is a very important central cellular organelle. On the other hand according to EV researchers extracellular vesicles are cellular organelles too.

10) 38 bacteria. As the name suggests, transport vesicles carry proteins and other molecules within the cell,

39 for example from endoplasmic reticulum to Golgi apparatus and vice versa or from one part of Golgi

40 to another [2].

Is it better to write a literature text with the phrases “as the name suggests’; the descriptions are not succinct as required for a reasonable scientific text.

Expression “As the name suggest” was removed from the text.

11) Table 1 is the best seen since the figures are not visible. How can the Authors explain that the vesicles enriched with plasma membrane and vacuolar ATPases?

Just the destruction of the cells and artificial formation of closed vesicles from plasma membrane and of vacuolar membrane.

Thank you for appreciating table 1 (Figure 1 has been revised).

We expected the samples to be enriched in plasma and vacuolar membrane ATPases as vesicle preparations (especially referring to Evs) are usually rich in membrane proteins respect to cytosolic ones, of which ATPases are the most abundant ones.

12) Please, provide comparison of the vesicles by your method with vesicles isolated under cold in any type of description (size, electrophoretic properties etc.).

The vesicles we isolated in the 1M sucrose cushion were expected to be less heterogeneous than the crude vesicle fraction usually isolated by differential ultracentrifugation (ref 13). Importantly, they were isolated with similar buoyant density as mammalian exosomes, so we expected to isolate exosome-like entities.

13) The Reviewer is sorry to say but the parts of the text are rather about nothing without clear description and statements. E.g., again just simple general phrases as below. Pls, brush up all the text making it clear and concentrated.

We have tried to improve this aspect in the revised version that indeed contains many modifications.

3.4. Proteins involved in molecular transport

18 Analysis of different types of vesicles shows the protein/lipid composition plays a crucial role in

19 their formation [17]. Membrane proteins (MPs), besides being an important structural element of the

20 vesicle membranes, are also responsible for carrying out specific functions such as cargo selection

21 and fusion and allowing the vesicle to interact with its environment in a controlled way. MPs that are

22 prevalently responsible for the interaction between vesicle and environment are transporters,

23 channels, enzymes, receptors, anchorage site and signal transductors. Many integral MPs are

24 transmembrane proteins (TM), which span the lipid bilayer with portions exposed on both sides of

25 the membrane. The presence of MPs such as tetraspanins [18], and aquaporins [19] has been described

26 in the various vesicle types.

14) The P-type ATPases are

49 known to form a phosphorylated intermediate state during their ion transport cycle [22]. They

maintain the electro-chemical gradients across cellular membranes, 50 by translocating cations, heavy

51 metals and lipids. In fruit juice cells of Citrus species, the vacuole can be very acidic (pH 2). Recent

52 work shows that for this hyper acidification a vacuolar proton-pumping P-ATPase complex could be

53 responsible [20].

Reference [22] is 210 22. Termini, C.M.; Gillette, J.M. Tetraspanins Function as Regulators of Cellular Signaling. Front.

211 Cell Dev. Biol. 2017, 5, 34–34.

The text is again simply too description and not precise.

We have tried our best to focus the descriptions.

15) Technically, numbering of lines starts from page 7 repeatedly, the same with page numbering.

We were unable to make the numbering continuous after the page break we introduced with the table.

16) 54 Plants have many more ATP Binding Cassette (ABC) transporters than mammalian cells. ABC

55 transporters are grouped into 8 subfamilies. In our samples, we have found 11 ABC transporter

56 subunits from 6 subfamilies A, B, C, E, F and G.

References?

Thank you for this suggestion. We have inserted a new reference on plant ABC transporters Plant ABC Transporters by Joohyun Kang,Jiyoung Park, Hyunju Choi,Bo Burla,b,Tobias Kretzschmar,b,* Youngsook Lee,a,c and Enrico Martinoiaa.

Morover, the ABC transporters we have identified are listed in Supp Table 3.

17) Aquaporins care channel proteins regulating the flux of

61 water and other various small solutes across membranes. Aquaporins are important not only for the

62 plant physiology in the maintenance of water homeostasis but also in the interaction between plant

63 and environment. Vesicles containing aquaporin are recently get into the focus of EV research.

Language is not good, pls, check all the text.

“Aquaporins care channel proteins…” is nonsense.

Interaction between plant and environment is not related to plant physiology?

We have corrected this error.

18) All the text with plenty of potentially good and valuable data should be completely revised and restructured/reconsidered. So far here is a very shallow approach.

We have made a major revision of the text.

Sincerely yours,

The authors